# Neural Action Policy Safety Verification: Applicablity Filtering

**Primary Keywords:** *None*

## Abstract

Neural networks (NN) are an increasingly important representation of action policies $\pi$. Applicability filtering is a commonly used practice in this context, restricting the action selection in $\pi$ to only applicable actions. Policy predicate abstraction (PPA) has recently been introduced to verify safety of neural $\pi$, through over-approximating the state space subgraph induced by $\pi$. Thus far however, PPA does not permit applicability filtering, which is challenging due to the additional constraints that need to be taken into account. Here we overcome that limitation, through a range of algorithmic enhancements. In our experiments, our enhancements achieve several orders of magnitude speed-up over a baseline implementation, bringing PPA with applicability filtering close to the performance of PPA without such filtering.

## 1 Introduction

Neural networks (NN) are an increasingly important representation of action policies in many contexts, including AI planning (Issakkimuthu, Fern, and Tadepalli 2018; Groshev et al. 2018; Garg, Bajpai, and Mausam 2019). But how to verify that such a *policy* $\pi$ is safe? Given a *start condition* $\phi_0$ and an *unsafety condition* $\phi_u$, how to verify whether a unsafe state $s^u \models \phi_u$ is reachable from a start state $s^0 \models \phi_0$ under $\pi$? Such verification is potentially very hard as it compounds the state space explosion problem with the difficulty of analyzing even single NN decision episodes. A prominent line of works addresses neural controllers of dynamical systems, where the NN output forms input to a continuous state-evolution function (Tran et al. 2019; Huang et al. 2019; Dutta, Chen, and Sankaranarayanan 2019; Ivanov et al. 2021). A recent thread explores bounded-length verification of neural controllers (Akintunde et al. 2018, 2019; Amir, Schapira, and Katz 2021).

Here we follow up on work on *policy predicate abstraction* (PPA) by Vinzent et al. (2022; 2023) (henceforth: *VEA*), which tackles neural policies $\pi$ that take discrete action choices in non-deterministic state spaces. Like classical predicate abstraction (Graf and Saïdi 1997), PPA builds an over-approximating abstraction defined through a set $\mathcal{P}$ of *predicates*, i.e., linear constraints over the state variables. However, PPA abstracts not the full state space, but the subgraph induced by $\pi$. To compute the abstract state space $\Theta_{\mathcal{P}}^{\pi}$, one must repeatedly solve the sub-problem of deciding whether there is a transition from abstract state $s_{\mathcal{P}}$ to abstract state $s_{\mathcal{P}}'$ under $\pi$. This *abstract transition problem* is encoded into satisfiability modulo theories (SMT) (Barrett and Tinelli 2018), and answered querying solvers tailored to NN analysis (Katz et al. 2019). If there does does not exist a path from $\phi_0$ to $\phi_u$ in $\Theta_{\mathcal{P}}^{\pi}$, then $\pi$ is safe. Counterexample-guided abstraction refinement (CEGAR) (Clarke et al. 2003) is deployed to iteratively refine $\mathcal{P}$ until either $\pi$ is proven safe or an unsafe counterexample is found.

VEA consider neural policies that may select any action in any state, including *inapplicable* actions. This makes it unnecessarily difficult to learn good policies. Instead an established practice is to *filter* the selection of $\pi$ with respect to *applicability* (Toyer et al. 2020; Stahlberg, Bonet, and Geffner 2022). On the verification side, however, applicability filtering is challenging since it introduces additional disjunctive behavior into the abstract transition problem: $\pi$ *may select action label $l$ depending on whether another action $l'$ is or is not applicable.* Implemented straightforwardly, PPA with applicability filtering suffers from a huge performance loss. In our experiments on VEA's benchmarks, it runs of ouf time or memory on all but the smallest instances − which, without applicability filtering, PPA tackles in a few seconds. In this paper, we devise a range of algorithmic enhancements that overcome this limitation. The enhancements exploit SMT-solver-specific encoding strategies, and simplify disjunctions in the SMT encoding of the applicability filter based on entailment of sub-constraints. Empirically, these methods achieve runtime improvements of up to three orders of magnitude, and bring PPA with applicability filtering close to the performance of PPA without such filtering.

## 2 Preliminaries

We consider discrete non-deterministic transition systems described by a tuple $\langle \mathcal{V}, \mathcal{L}, \mathcal{O} \rangle$ where $\mathcal{V}$ is a finite set of bounded-integer *state variables*, $\mathcal{L}$ is a finite set of *action labels* and $\mathcal{O}$ is a finite set of *operators*. We denote by *Exp* the set of *linear expressions* over $\mathcal{V}$, i.e., of the form $\sum_{v \in \mathcal{V}} d_v \cdot v + c$ with coefficients $d_v \in \mathbb{Z}$ for each $v \in \mathcal{V}$ and $c \in \mathbb{Z}$. Accordingly, $C$ denotes the set of *linear constraints*, of the form $\sum_{v \in \mathcal{V}} d_v \cdot v \geq c$, and Boolean combinations thereof. An *operator* $o \in \mathcal{O}$ is a tuple $(l, g, u)$ with *label* $l \in \mathcal{L}$, *guard*

$g \in C$ (a conjunction of linear constraints), and (linear) *update* $u\colon \mathcal{V} \to Exp$.

The *state space* of $\langle \mathcal{V}, \mathcal{L}, \mathcal{O} \rangle$ is a labeled transition system $\Theta = \langle \mathcal{S}, \mathcal{L}, \mathcal{T} \rangle$. The set of *states* $\mathcal{S}$ is the finite set of all complete variable assignments over $\mathcal{V}$. The set of *transitions* $\mathcal{T} \subseteq \mathcal{S} \times \mathcal{L} \times \mathcal{S}$ contains $(s, l, s')$ iff there exists an operator $o = (l, g, u)$ such that $g$ is satisfied over $s$, also written $s \models o$, and $s'(v)$ maps to the update $u(v)$ evaluated over $s$ for each $v \in \mathcal{V}$, formally $s' = \{v \mapsto u(v)(s) \mid v \in \mathcal{V}\}$, also abbreviated $s' = s[\![o]\!]$.

An *action policy* $\pi$ is a function $\mathcal{S} \to \mathcal{L}$. We consider $\pi$ represented by a *neural network* (NN). Specifically, we focus on feed-forward NN with *rectified linear unit* (ReLU) activations $ReLU(x) = \max(x, 0)$. These NN consist of an input layer, arbitrarily many hidden layers, and an output layer with one neuron per label $l \in \mathcal{L}$. A *safety property* is a pair $(\phi_0, \phi_u)$, where $\phi_0 \in C$ and $\phi_u \in C$ identify the set of start and unsafe states respectively. A policy $\pi$ is *unsafe* with respect to $(\phi_0, \phi_u)$ iff there exists a state path $\langle s^0, \dots, s^n \rangle$ such that $s^0 \models \phi_0$, $s^n \models \phi_u$, and $(s^i, \pi(s^i), s^{i+1}) \in \mathcal{T}$ for $i \in \{0, \dots, n-1\}$. Otherwise $\pi$ is *safe*.

**Policy predicate abstraction** (PPA) (Vinzent, Steinmetz, and Hoffmann 2022) is an extension of classical predicate abstraction (Graf and Saïdi 1997). Unlike its classical counterpart, PPA abstracts not the full state space, but the subgraph induced by $\pi$. Assume a set of *predicates* $\mathcal{P} \subseteq C$. An *abstract state* $s_{\mathcal{P}}$ is a complete truth value assignment over $\mathcal{P}$. $[s_{\mathcal{P}}] = \{s \in \mathcal{S} \mid \forall p \in \mathcal{P}\colon p(s) = s_{\mathcal{P}}(p)\}$ denotes the set of concrete states represented by $s_{\mathcal{P}}$. The *policy predicate abstraction* of $\Theta$ over $\mathcal{P}$ and $\pi$ is the labeled transition system $\Theta_{\mathcal{P}}^{\pi} = \langle \mathcal{S}_{\mathcal{P}}, \mathcal{L}, \mathcal{T}_{\mathcal{P}}^{\pi} \rangle$ where $\mathcal{S}_{\mathcal{P}}$ is the set of abstract states over $\mathcal{P}$ and $(s_{\mathcal{P}}, l, s'_{\mathcal{P}}) \in \mathcal{T}_{\mathcal{P}}^{\pi}$ iff there exists $(s, l, s') \in \mathcal{T}$ such that $s \in [s_{\mathcal{P}}]$, $s' \in [s'_{\mathcal{P}}]$ and $\pi(s) = l$.

Analogously to safety in $\Theta$, $\pi$ is said to be *unsafe* in $\Theta_{\mathcal{P}}^{\pi}$ iff there exists an abstract path $\langle s_{\mathcal{P}}^0, l^0, \dots, l^{n-1}, s_{\mathcal{P}}^n \rangle$ such that $s^0 \models \phi_0$ for some $s^0 \in [s_{\mathcal{P}}^0]$, $s^n \models \phi_u$ for some $s^n \in [s_{\mathcal{P}}^n]$, and $(s_{\mathcal{P}}^i, l^i, s_{\mathcal{P}}^{i+1}) \in \mathcal{T}_{\mathcal{P}}^{\pi}$ for $i \in \{0, \dots, n-1\}$. Otherwise $\pi$ is *safe* in $\Theta_{\mathcal{P}}^{\pi}$, in which case it is safe in $\Theta$ as well. An (unsafe) abstract path in $\Theta_{\mathcal{P}}^{\pi}$ may be *spurious*, i.e., there does not exist a corresponding path in $\Theta$ under $\pi$. *Counterexample-guided abstraction refinement* (CEGAR) (Clarke et al. 2003) iteratively removes such spurious abstract paths by refining $\mathcal{P}$, until either the abstraction is proven safe, or a non-spurious abstract path is found proving $\pi$ unsafe. VEA provide a CEGAR framework specialized to PPA (Vinzent, Sharma, and Hoffmann 2023).

To compute $\Theta_{\mathcal{P}}^{\pi}$, one must solve the **abstract transition problem** for every possible abstract transition: $(s_{\mathcal{P}}, l, s'_{\mathcal{P}}) \in \mathcal{T}_{\mathcal{P}}^{\pi}$ iff for some $l$-labeled operator $o \in \mathcal{O}$ there exists a concrete state $s \in [s_{\mathcal{P}}]$ such that $s \models o$, $s[\![o]\!] \in [s'_{\mathcal{P}}]$ and $\pi(s) = l$. In the classical setting where no policy is considered and thus condition $\pi(s) = l$ is not needed, such abstract transition problems are routinely encoded into satisfiability modulo theories (SMT) (e.g. (Barrett and Tinelli 2018)). For PPA however, the policy condition $\pi(s) = l$ introduces a key new source of complexity as the SMT sub-formula representing the neural network $\pi$ contains one non-linear constraint for every ReLU activation. VEA show how this can be dealt with through approximate SMT checks. In particu-

lar, they use continuous relaxations of the bounded-integer state variables, which can be dispatched to *Marabou* (Katz et al. 2019), an SMT solver tailored to NN analysis.

## 3 Applicability Filtering

VEA consider neural action policies that are obtained by applying argmax to the output of the NN. Let $\pi_l(s)$ be the NN output for label $l \in \mathcal{L}$ given input state $s \in \mathcal{S}$, then $\pi(s) = \operatorname{argmax}_{l \in \mathcal{L}} \pi_l(s)$. Such $\pi$ may select any label in any state, even if it is not *applicable*, i.e., there does not exist $s' \in \mathcal{S}$ such that $(s, l, s') \in \mathcal{T}$, or equivalently, there does not exist an $l$-labeled operator $o$ with $s \models o$.

From a learning perspective, allowing $\pi$ to select inapplicable actions is unnecessarily difficult, as $\pi$ must learn which actions are applicable in which state. A simple commonplace technique to avoid this is to *filter* the selection of $\pi$ with respect to *applicability* (e.g. (Toyer et al. 2020)). Formally, the policy under applicability filtering is defined

$$\pi(s) = \operatorname*{argmax}_{\{l \in \mathcal{L} \mid \exists o \in \mathcal{O}_l\colon s \models o\}} \pi_l(s)$$

where $\mathcal{O}_l = \{(l, g, u) \in \mathcal{O}\}$ is the set of $l$-labeled operators.

From a verification perspective, applicability filtering also is desirable because, without such filtering, a policy run may be safe simply because of *stalling*, selecting an inapplicable action which ends the run.

However, applicability filtering adds an additional source of complexity to the abstract transition problem, specifically to the policy condition $\pi(s) = l$. In what follows, we focus on the SMT encoding of this sub-problem. The encoding of the neural network itself remains unaffected; we provide a full specification of the SMT encoding in the appendix.

Let $\pi_l$ denote the SMT variable representing the NN output of label $l$. Without filtering, the policy selection condition is a simple conjunction $\bigwedge_{l' \in \mathcal{L} \setminus \{l\}} \pi_l > \pi_{l'}$. Under applicability filtering however, each conjunct here becomes a disjunction $\bigwedge_{l' \in \mathcal{L} \setminus \{l\}} (\pi_l > \pi_{l'} \vee \neg \bigvee_{o \in \mathcal{O}_{l'}} g_o)$ where $g_o$ denotes the guard of operator $o$. In words: either the output value of $l$ is greater than that of $l'$, or $l'$ is not applicable. Since each $g_o$ is a conjunction of linear constraints, the selection condition expands to

$$\bigwedge_{l' \in \mathcal{L} \setminus \{l\}} \left( \pi_l > \pi_{l'} \vee \neg \bigvee_{o \in \mathcal{O}_{l'}} \bigwedge_{i \in \{1, \dots, m\}} g_o^i \right)$$

where *sub-guard* $g_o^i$ denotes the $i$-th linear constraint of guard conjunction $g_o$ and $m$ is the guard size.[1]

## 4 Enhancements

Applicability filtering extends the SMT encoding of the abstract transition problem by a layer of convoluted disjunctions. To tackle this new source of complexity, we devise a

---

[1]To simplify notation, we assume $m$ constant over all guards. One can extend any guard to some maximal $m$ by adding *trivially-true* constraints.

range of encoding enhancements that target disjunctions in general and the applicability filter in particular.

**Per-operator disjunctions.** One type of enhancements exploits the way disjunctions are encoded in *Marabou*, the NN-tailored SMT solver underlying VEA's algorithm. *Marabou* supports disjunctions in disjunctive normal form (DNF), i.e., $\bigvee_i \bigwedge_j \phi_i^j$ with linear constraints $\phi_i^j$. Naively rewriting the top-disjunction $\pi_l > \pi_{l'} \vee \neg \bigvee_{o \in \mathcal{O}_{l'}} \bigwedge_i g_o^i$ into DNF one obtains $\pi_l > \pi_{l'} \vee \bigwedge_{o \in \mathcal{O}_{l'}} \bigvee_i \neg g_o^i$ and then

$$\pi_l > \pi_{l'} \vee \bigvee_{f \in (\mathcal{O}_{l'} \to \{1,\ldots,m\})} \bigwedge_{o \in \mathcal{O}_{l'}} \neg g_o^{f(o)}$$

where $\mathcal{O}_{l'} \to \{1, \ldots, m\}$ is the set of sub-guard combinations over $\mathcal{O}_{l'}$, Since there are $m^{|\mathcal{O}_{l'}|}$ combinations in total, this encoding is prone to result in a blow-up in size. We overcome this scalability issues by an alternative encoding that splits the top-disjunction into smaller disjunctions

$$\pi_l > \pi_{l'} \vee \bigvee_{i \in \{1,\ldots,m\}} \neg g_o^i$$

one for each operator $o \in \mathcal{O}_{l'}$ (PER-OP-DISJ).

**Reusing slack variables.** *Marabou* transforms every disjunction $\phi$ to only contain bound tightenings $v \geq c$. Specifically, every non-bound constraint $\sum_{v \in \mathcal{V}} d_v \cdot v \geq c$ in $\phi$ is transformed to an equation $\sum_{v \in \mathcal{V}} d_v \cdot v + a = c$ where $a$ is a fresh slack variable. This transformed equation is added to the global encoding in a conjunctive manner. The constraint in $\phi$ is replaced by a bound tightening $a \leq 0$.

We optimize this transformation in that we check for multiple occurrences of constraints (identical up to variable re-ordering) over all disjunctions (OPT-SLACK-VAR). For each re-occurring constraint, we introduce only a single slack variable and add the transformed equation only once to the global encoding. In particular, this pertains to PER-OP-DISJ where $\pi_l > \pi_{l'}$ occurs multiple times.

**Entailed sub-constraints.** Another type of enhancements exploits *entailment* to simplify the encoding. Given constraints $\phi, \psi \in C$, we say $\phi$ *entails* $\psi$, written $\phi \vdash \psi$, iff for every assignment $s \in \mathcal{S}$ such that $s \models \phi$ it also holds $s \models \psi$. Let $\bigvee_i \bigwedge_j \phi_i^j$ be a disjunction contained in $\phi$. If, for some $i$ and $j$, $\phi \vdash \phi_i^j$, then $\phi_i^j$ can be removed. If, for some $i$, $\phi \vdash \phi_i^j$ for every $j$, then $\phi$ entails disjunct $i$ and so the entire disjunction, which can be removed from $\phi$. If $\phi \vdash \neg \phi_i^j$, then the entire disjunct $i$ is infeasible and can be removed. If all disjuncts $i$ are infeasible, then the entire disjunction is infeasible and so is $\phi$. We apply entailment information to optimize the encoding of disjunctions on two levels.

Firstly, on a per operator level (ENTAIL-OP). For each operator $o$, VEA's algorithm to compute $\Theta_{\mathcal{P}}^\pi$ runs an *applicability test* $\exists s \in [s_{\mathcal{P}}]: s \models o$. If this test fails then the guard conjunction $g_o$ is entailed to be infeasible in abstract state $s_{\mathcal{P}}$. Say $o$ is $l'$-labeled. We can use this entailment information to simplify the policy condition for any label $l \neq l'$.

Secondly, on a generic linear level (ENTAIL-GEN) with entailed $\psi$ in the form of a linear constraint $\sum_{v \in \mathcal{V}} d_v \cdot v \geq c$.

Let $lo_v(\phi)$ and $up_v(\phi)$ denote a lower and upper bound for $v$ entailed by $\phi$ respectively. Then $\phi$ entails $\psi$ if

$$\sum_{v \in \mathcal{V}^+} d_v \cdot lo_v(\phi) + \sum_{v \in \mathcal{V}^-} d_v \cdot up_v(\phi) \geq c$$

where $\mathcal{V}^+ = \{v \in \mathcal{V} \mid d > 0\}$ denotes the variable set with positive coefficients, and $\mathcal{V}^- = \{v \in \mathcal{V} \mid d < 0\}$ the variable set with negative coefficients. Analogously, $\phi$ entails $\neg \psi$, we also say $\psi$ is *infeasible*, if

$$\sum_{v \in \mathcal{V}^+} d_v \cdot up_v(\phi) + \sum_{v \in \mathcal{V}^-} d_v \cdot lo_v(\phi) < c.$$

$\phi$ in the form of the abstract transition problem *syntactically* entails variable bounds in that many predicates in $\mathcal{P}$ are bound constraints $v \geq c$ and, thereby, the conditions $s \in [s_{\mathcal{P}}]$ and $s[\![o]\!] \in [s'_{\mathcal{P}}]$ involve bound tightenings. In addition, Marabou deploys techniques to derive tight bounds on the NN outputs (e.g., (Singh et al. 2019)).

# 5 Experiments

We implemented our approach on top of VEA's C++ code base. The enhancements are mostly implemented directly into *Marabou*, in particular OPT-SLACK-VAR and ENTAIL-GEN (which is a contribution to improve Marabou's performance on disjunctions in general).[2] All experiments were run on machines with Intel Xenon E5-2650 processors at 2.2 GHz, with time and memory limits of 12 h and 4 GB.

**Benchmarks.** We use VEA's benchmarks. These are non-deterministic variants of the planning domains Blocksworld, SlidingTiles and Transport encoded in JANI (Budde et al. 2017). For each domain instance, there are three NN policies trained by VEA using Q-learning (Mnih et al. 2015), each with two hidden layers of size 16, 32 and 64 respectively, and with ReLU activation nodes. There are policies that do, and ones that do not, take move costs into account.

The policies by VEA are trained without applicability filtering. In our evaluation, we verify these same policies with and without applicability filtering, to allow direct comparison of verification performance.

**Configurations.** We compare a range of algorithmic configurations combining different applicability filter enhancements for abstract transition computation as part of VEA's verification algorithm.

- NoOpt disables and AllOpts enables all enhancements.
- OnlyPerOp, OnlySlack, OnlyOp, OnlyGen only enables PER-OP-DISJ, OPT-SLACK-VAR, ENTAIL-OP, ENTAIL-GEN respectively.
- NoPerOp, NoSlack, NoOp, NoGen enables all enhancements except PER-OP-DISJ, OPT-SLACK-VAR, ENTAIL-OP, ENTAIL-GEN respectively.
- NoApp verifies the policy without applicability filtering, as done by VEA.

---

[2]All our source code (tool and experiments) will be made publicly available upon publication.

| Benchmark | NN | Safe | Time | | | | | | | | | | NoApp | |
|---|---|---|---|---|---|---|---|---|---|---|---|---|---|---|
| | | | NoOpt | OnlyPerOp | OnlySlack | OnlyOp | OnlyGen | NoPerOp | NoSlack | NoOp | NoGen | AllOpts | Safe | Time |
| 4 Blocks (cost-ign) | 16 | ✓ | 8797 | 31 | 8712 | 32 | 24 | 23 | **21** | 22 | 24 | 22 | ✓ | 6 |
| | 32 | ✓ | 18889 | 115 | 16850 | 108 | 49 | 54 | 40 | 40 | 79 | **38** | ✓ | 10 |
| | 64 | ✓ | 29275 | 1618 | 28595 | 1296 | 259 | 255 | 235 | **221** | 1241 | 228 | ✓ | 16 |
| 6 Blocks (cost-ign) | 16 | ✓ | - | - | - | - | - | - | **27350** | 30929 | 28358 | 27457 | ✓ | 124 |
| | 32 | ✓ | - | 29866 | - | - | - | - | 9817 | 9059 | 12272 | **9037** | ✓ | 81 |
| | 64 | ? | - | - | - | - | - | - | - | - | - | - | ✓ | 631 |
| 8 Blocks (cost-ign) | 16 | ? | - | - | - | - | - | - | - | - | - | - | ✓ | 9593 |
| | 32 | ? | - | - | - | - | - | - | - | - | - | - | ? | - |
| | 64 | ? | - | - | - | - | - | - | - | - | - | - | ? | - |
| 8-puzzle (cost-ign) | 16 | × | - | 129 | 1387 | - | 246 | 116 | 103 | 80 | 87 | **77** | × | 44 |
| | 32 | × | - | 14385 | - | - | 16674 | 13963 | 13356 | 13024 | 13012 | **12610** | ✓ | 16727 |
| | 64 | × | - | 17417 | - | - | 17542 | 15056 | 16151 | 11729 | 11638 | **11453** | ? | - |
| 4 Blocks (cost-awa) | 16 | ✓ | - | 159 | 41442 | 126 | 58 | 53 | 52 | 53 | 94 | **49** | ✓ | 36 |
| | 32 | ✓ | - | 4188 | - | 3171 | 483 | 454 | 491 | **449** | 2500 | 454 | ✓ | 329 |
| | 64 | ? | - | - | - | - | - | - | - | - | - | - | ✓ | 36214 |
| 6 Blocks (cost-awa) | 16 | ✓ | - | - | - | - | - | - | 29842 | 30083 | 36635 | **28262** | ✓ | 8992 |
| | 32 | ? | - | - | - | - | - | - | - | - | - | - | ✓ | 27215 |
| | 64 | ? | - | - | - | - | - | - | - | - | - | - | ? | - |
| 8 Blocks (cost-awa) | 16 | × | - | 1460 | - | - | - | - | 909 | **866** | 929 | 915 | × | 295 |
| | 32 | ? | - | - | - | - | - | - | - | - | - | - | ? | - |
| | 64 | ? | - | - | - | - | - | - | - | - | - | - | ? | - |
| 8-puzzle (cost-awa) | 16 | × | - | 2806 | - | - | 6143 | 2697 | 2420 | 2049 | 2010 | **1977** | × | 2881 |
| | 32 | × | - | 12908 | - | - | 13698 | 11931 | 11478 | 11458 | 11267 | **10917** | ? | - |
| | 64 | × | - | - | - | - | 39354 | 35643 | 35922 | 30522 | 33109 | **29331** | ? | - |
| Transport | 16 | × | 23 | **0.4** | 23 | 23 | 12 | 10 | **0.4** | 0.5 | **0.4** | 0.5 | × | 0.3 |
| | 32 | × | 37 | **0.5** | 494 | 36 | 12 | 12 | **0.5** | 1 | 1 | 1 | × | 0.4 |
| | 64 | × | 28 | **1** | 53 | 28 | 19 | 17 | **1** | **1** | **1** | **1** | × | 1 |

Table 1: Runtime results in seconds for the evaluated configurations of enhancements for applicability filtering over different benchmarks and NN policies. - indicates runs that exceed the resource limit of 12h time and 4 GB memory.

**With vs. without enhancements.** Table 1 shows our results. AllOpts clearly dominates NoOpt. The latter only terminates on the smallest problem instances, with a runtime offset of up to three orders of magnitude.

**Ablation study.** OnlyPerOp covers 9 additional instances compared to NoOpt. This indicates that the choice of encoding (PER-OP-DISJ or not) is a crucial factor for efficiency. That said, also the other configurations with a single enhancement, especially OnlyGen, increase coverage compared to NoOpt. Moreover, AllOpts outperforms every single-enhancement configuration and always covers additional instances. This shows that also the combination of enhancements is crucial.

NoPerOp performs competitive on smaller Blocksworld instances, but fails on larger ones similar to NoOpt. On 8-puzzle and Transport it performs consistently slower than AllOpts. Again, this demonstrates the relevance of PER-OP-DISJ. NoOp tends to be more efficient than NoGen. This indicates that ENTAIL-GEN is more crucial than ENTAIL-OP. NoSlack performs particularly good on Blocksworld but particularly bad on some 8-puzzle problem instances. On the former domain operator guards contain many bound constraints. This shows that OPT-SLACK-VAR becomes more relevant when operator guards are more complex.

**Applicability-Filtering vs. No-Filtering.** Clearly, the additional complexity of applicability filtering in SMT can increase verification time. On Blocksworld, AllOpts is worse than NoApp, covering four instances less. On 8-puzzle, on the other hand, AllOpts covers three more instances than NoApp and is competitive on the remaining ones. This is presumably due to the actual verification *results* – on NN 32 (cost-ign), the policy is safe without applicability filtering, but is unsafe with applicability filtering. This exemplifies that, without applicability filtering, a policy may be safe due to stalling. This questionable form of safety is no longer possible under applicability filtering. Presumably, the same issue occurs in the 8-puzzle instances not covered by NoApp. On Blocksworld and Transport, there are no such verification result differences. In particular, on the former all policies are safe with and without applicability filtering.

# 6 Conclusion

The verification of neural action policies is important. Here we contribute enhancements for PPA with applicability filtering, getting rid of much of the additional complexity suffered by a baseline implementation.

Important future directions for PPA include liveness properties, in particular the guarantee that a policy will eventually reach the goal; partial safety verification, continuing CEGAR on instances already proved to be unsafe, in order to identify safe regions of the state space; and the extension to probabilistic and/or continuous-state transition systems. Our enhancements are orthogonal to all these extension.

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
