# OpenReview forum: "Neural Action Policy Safety Verification: Applicablity Filtering"
_icaps-conference.org/ICAPS/2024/Conference — ICAPS 2024_

### Official Review · Reviewer_jeun · 2023-12-25

**Significance And Importance:** 2
**Soundness:** 4
**Novelty:** 2
**Clarity:** 3
**Confidence:** 3

**Weaknesses:**

-1: Major weaknesses requiring significant work to be addressed for the paper to be accepted.

**Contributions Of The Paper:**

This paper presents an improvement to a technique that combines CEGAR and SMT for the verification of policies represented by a neural network.  Specifically, given a non-deterministic planning instance, given a policy that has been learned for this instance and that is represented by a neural network, the goal is to determine whether this policy may lead the environment in a specific set of states labelled as a unsafe.  This is done via a CEGAR/IC3 method in which the transition system is abstracted with "super states" that share similar properties: if this abstracted transition system admits no path to an unsafe state, then the policy is safe; otherwise, the system is refined until a fully refined path is found that leads to the unsafe state.  Importantly, the transitions of the abstract transition system are tested through an SMT query that decides whether there exist two states $s$ and $s'$ in the two abstract states such that applying the action in $s$ can lead to $s'$.

My understanding is that the existing work assumes that all actions are applicable in all states (or, at least, the SMT query does not check whether the action is applicable in the state $s$).  The authors show how to modify the SMT query to add this check; there is also a long discussion on how to reformulate the query and to perform preprocessing steps that accelerate the SMT query.  The proposed changes are verified experimentally with improvements up to multiple orders of magnitude.

**Ethical Considerations:**

(4) Good: The paper adequately addresses most, but not all, of the applicable ethical considerations

**Nomination For Best Paper:**

No

**Overall Evaluation:**

-1: (weak reject)

**Questions For Authors:**

I do not have questions about the contributions of the paper as they are clearly explained.  My issues are primarily about how you implemented the improvements in practice, but I would like to see answers *in the paper*, not just as rebuttal answers.  Instead, my question is: Could you justify why this paper should be accepted as an ICAPS short paper rather than in another venue where you might provide more details?

**Reproducibility:**

4: Authors promise to release code and domains (whichever apply).

**Strengths Of The Paper:**

The paper is extremely well-written.  The authors did a great job to provide all the necessary definitions in the small number of pages allowed by a short paper submission.  The paper is clear and well motivated.  The results are also very significant with several orders of magnitude improvement.

**Weaknesses Of The Paper:**

There are two potential issues with the paper which are: 1. the contribution and 2. the appropriateness of the venue/format.

The contribution is arguably relatively small.  Indeed, the paper essentially presents some "small" tricks to simplify SMT formulas.  I understand that these tricks do have a huge impact, so I would be willing to consider the paper acceptable in this regard.

While the paper is very well-written, there is an argument that 4 pages is too few to understand the content.  The format does not allow the authors to explain the meaning of the definitions and provide examples.  This is arguably not an issue because the reader is probably assumed to have a sufficient understanding of the problem (and to know such papers as VEA).  I suspect that the authors considered that the contribution is not significant enough to deserve a full paper.  As it stands, I feel that I did not understand how and when these tricks are used.

In particular, I expect that the entailments carry from SMT problems to SMT problems.  For instance, if one tests whether there a transition from an abstract state $S$ to another abstract state $S'$ labelled with a certain action, and then to another abstract state $S'_2$, some (all?) of the entailments can be kept.  If this state $S'$ is later split, the entailments should still be applicable.  How is this information stored and reused?

I suggest that the authors publish this work in another venue where they will be able to give more details about their work and their implementation.  An ICAPS workshop or a technical report seem a better fit.

(I add my comments/suggestions about the paper below.  Notice that they are not "Weaknesses" per se; the OpenReview form does not permit this type of feedback.)

-

I understand that space is very tight in the paper, but would it be possible to explain that non-determinism stems from the fact that there may be multiple operators for the same label?

Being overly pedantic (and people might disagree with me): in the introduction, the $u$ is $s^u$ is used to indicate that the state is unsafe while $n$ in $s^n$ is the step of in the sequence of states.  Ideally, I would like to see a different notation, but there may be no satisfactory solution.

For $p$ a predicate and $s$ a state, $p(s)$ is not defined.
$s \models \phi$ is used to indicate that $s$ satisfies the constraint $phi$ (e.g., Line 103).  Would it be better to use $\phi(s)$ to be consistent with $p(s)$?

The notation $[s_P]$ is used to represent the set of states that are consistent with the abstract state $s_P$, while $s\llbracket o\rrbacket$ is used to represent the effect of $o$.  It is my understand that some communities such as Formal Methods use $\llbracket x \rrbracket$ to refer to what $x$ represents.  Should you swap the two notations?

It is unclear whether the word "spurious" include the following scenario: there does exist a path in the original LTS that corresponds to the path in the abstract LTS, but this path either does not start in an initial state or does not end in an unsafe state.

Technically speaking, one cannot say that CEGAR removes spurious paths because the paths in the one-step-refined LTS are different (in other words, not a subset) of the paths in the LTS at the previous iteration.

The end of Section 2 is underwhelming.  Do I understand correctly that, since VEA uses approximation, the CEGAR approach is not guaranteed to terminate?  Or does the approximation disappear as the abstract states become closer and closer to concrete states?

Between Lines 150 and 155, "there is no transition" is equated to "there does not exist an $l$-labelled operator $o$ with $s \models o$.  Technically, this was never mentioned before.  According to earlier definitions, it could be that $s \models o$ but $s\llbracket o\rrbracket$ falls outside of $S$.

***** "From a learning perspective, allowing $\pi$ to select inapplicable actions is unnecessarily difficult [...]"
Did you mean the opposite?

If you really need to save a couple of lines, I think you can use the notation $m(g)$ for the size of guard $g$ (which removes the need for the footnote).

The equation of Line 173 looks impressive, but note that it can be rewritten
  big-conjunction-over-l'-and-o
    (pi_{l'} >= pi_l) => \bigvee_i \not g_o^i
which looks slightly less impressive.

Line 189, should it be $a \ge 0$?

The explanation starting in Line 201 is very unclear.  Is it the case that
  $\phi = \bigvee_i\bigwedge_j \phi_i^j$?
Is that what you mean by "disjunction contained in $\phi$?  What does "$\phi_i^j$ can be removed" mean?

Misc.

Line 47
"If there does does not exist": Repeated "does".

Line 64:
"runs of ouf time"

Line 306:
"extension" -> extensions

---

> ### Author Rebuttal · Authors · 2024-01-26
>
> Thanks for the review.
>
> Regarding write-up/space: We can publish an online report with additional details if prompted by the reviewers; if preferred (and if offered by the conference), we can also buy an additional page for a final version. Regarding the question whether a different venue would be more suitable: as the reviewers have noted, the contribution of the paper is rather small at a technical novelty level; but the empirical benefits are significant. We believe that this is an ideal profile for an ICAPS short paper (quote CFP: "Short papers may be of narrower scope, for example, by ... proposing or evaluating a small, yet important, extension of previous work or new idea."). An online report or additional page should address the density problem well. For a full paper in any major AI conference, the authors' own judgement is that the paper would not be suitable (and would not be accepted).
>
> Response to additional comments (we will do our best to address all your suggestions in a final paper):
>
> Our implementation exploits the incremental nature of the abstract state expansion algorithm provided by VEA (Vinzent et al. 2022) to reuse entailment information between abstract transition checks. This algorithm constructs the SMT encoding incrementally, such that, e.g, ENTAIL-OP, is applied once per abstract source state and label and reused for all successor candidates. Entailment information is not cached between CEGAR iterations.
>
> VEA's spuriousness check include the start and unsafety constraints.
>
> VEA's CEGAR approach is sound and complete. They use approximate SMT-checks as part of solving the exact transition problem.
>
> 150-155: Here, we talk about transitions in the concrete state space, not the abstraction where s[[o]] may "fall outside S".
>
> 159: Applicability filtering facilitates training since the policy must only learn to select from the applicable actions rather than all actions.
>
> 189: If a <= 0, the transformed equation holds iff, as required, sum_{v in V} d_v * v >= c.
>
> 201: A disjunction is "contained" in phi if it occurs as sub-constraint -- in our case as a conjunct in the abstract transition problem encoding. "Removing phi_i^j" refers to simplifying disjunct i (by removing phi_i^j).

---

### Official Review · Reviewer_Ak4g · 2024-01-06

**Significance And Importance:** 1
**Soundness:** 3
**Novelty:** 2
**Clarity:** 4
**Overall Evaluation:** 1
**Confidence:** 3

**Weaknesses:**

1: Minor weaknesses that are easily fixable.

**Contributions Of The Paper:**

The paper extends an SMT encoding for safety verification of policies encoded in neural networks. The extension allows the verifier to only consider applicable actions in each state. The straightforward implementation of this extension incurs a large runtime penalty, however, so the authors propose several optimizations that reduce the runtime overhead. They evaluate these optimizations in a detailed ablation study.

**Ethical Considerations:**

(1) Not Applicable: The paper does not have any ethical considerations to address

**Nomination For Best Paper:**

No

**Questions For Authors:**

1) Both the AllOpts and NoApp variants fail to verify many policies within 12 hours, even for the smallest network size. How do you think we can scale predicate abstraction to more complicated policies?
2) To better justify the relevance of applicability filtering, shouldn't one quantify the effect of learning policies with vs. without applicability filtering? (I.e., how much does the applicability filtering help in the learning phase?)

**Reproducibility:**

4: Authors promise to release code and domains (whichever apply).

**Strengths Of The Paper:**

The paper addresses a relevant gap in the literature. It is well-written and easy to follow. The authors provide a detailed description of the SMT encoding and the optimizations. The ablation study is thorough and provides a good understanding of the effect of each optimization. The optimizations indeed reduce the runtime overhead significantly.

**Weaknesses Of The Paper:**

The enhancements are mostly part of the external verification tool (Marabou). As such it feels a bit strange to have an ICAPS paper about them, whereas we probably wouldn't publish a dedicated paper when the Madagascar planner switches to an improved underlying SAT solver or when Fast Downward switches to a CPLEX version that solves LPs faster.

Minor comments:

In Table 1, it would be nice to indicate which of the "--" entries is a timeout and which ran out of memory.

21: a --> an
26: works --> work
25-32: unclear how this relates to the text above and below it.
footnote 1: assume that m is constant
scalability issue--s--
197: enhancement--s--
caption of Table 1: explain that awa means "aware" or use "costs" and "no-costs" instead of "cost-awa" and "cost-ign".
275: good --> well, bad --> badly

---

> ### Author Rebuttal · Authors · 2024-01-25
>
> Thanks for the review.
>
> Concerning your remark on enhancements to Marabou in an ICAPS paper:
> The enhancements we devise here pertain to SMT encodings for Marabou in our context. Some of this is very specific to that context (PER-OP-DISJ, ENTAIL-OP) and hence naturally belongs into a paper about policy verification. The other enhancements are more generic (OPT-SLACK-VAR, ENTAIL-GEN), and could in principle be viewed as contributions to Marabou. However, also these enhancements are motivated by, and presumably mainly useful for, our specific context where the addressed phenomena frequently occur. In any case, these enhancements are small and hardly of value as a general contribution. Their value lies in the performance improvements for policy verification with applicability filtering, which we demonstrate here. We believe that this is perfectly suitable for an ICAPS short paper.
>
> 1) Much can still be done to improve scalability. PPA can benefit from any progress on NN analysis for single input-output episodes e.g., [1]. There is much potential for parallelization during the computation of abstract transitions. Furthermore, given that many abstract transition problems are similar in nature, one might exploit incrementality between calls to the SMT-solver. Finally, recent research [2] shows that also small NN (same size like the smallest NN considered in our paper) can be used in real-world robotic scenarios.
>
> 2) In our humble opinion, the motivation for applicability-filtering is very clear and strong -- it is the simplest possible method to ensure that a policy execution never selects inapplicable actions. We are not aware of any work on policies in planning that does not use applicability-filtering in some form. Yes, applicability-filtering may also affect the learning process itself. We have not explored that here as it is not the focus of this contribution, and in order to foster comparability to verification of policies without applicability filtering.
>
> [1] Shiqi Wang, Huan Zhang, Kaidi Xu, Xue Lin, Suman Jana, Cho-Jui Hsieh and J. Zico Kolter, Beta-CROWN: Efficient Bound Propagation with Per-neuron Split Constraints for Neural Network Robustness Verification, Neurips, 2021.
>
> [2] Guy Amir, Davide Corsi, Raz Yerushalmi, Luca Marzari, David Harel, Alessandro Farinelli, Guy Katz, Verifying learning-based robotic navigation systems, TACAS, 2023.

---

### Official Review · Reviewer_FsbX · 2024-01-14

**Significance And Importance:** 2
**Soundness:** 3
**Novelty:** 3
**Clarity:** 4
**Overall Evaluation:** 2
**Confidence:** 3

**Weaknesses:**

1: Minor weaknesses that are easily fixable.

**Contributions Of The Paper:**

This is a short follow up on two recent published studies about policy predicate abstraction. Contribution is a performance improvement via representation optimisations, and is specific to SMT-based bounded model checking safety properties of network-based policies (as can be derived using  Q-learning). In execution and in verification, network-based policies should be interpreted so that only applicable actions are executed. The paper deals with performance of verification with such applicability testing active when interpreting learnt policies -- i.e., verification with faithful execution semantics.

Model checking is done using the SMT solver Marabou, and contributions (three representational enhancements) are understood to be implemented in Marabou. Enhancements appear principled and correct, and are derived apparently from first principles and not by appealing to the model checking literature. SMT representational enhancements are based on: (i) formula rewriting of disjunctive constraints, as occur when faithfully representing the policy-network execution semantics in an SMT query, (ii) exploiting predicate subsumption (in manuscript 'entailment', but I think invoking a proof theory concept, such as entailment, for the purposes of this manuscript is not ideal), and (iii) slack variable re-use. Some empirical ablation work is provided, which is sufficient given the nature of this work.

Experimentation is on 9 problem instances in a convenient format, from the JANI bundle. Specifically, the experimental domains are: Tile, Blocks (e.g., upto 8), Transport.

**Ethical Considerations:**

(1) Not Applicable: The paper does not have any ethical considerations to address

**Nomination For Best Paper:**

No

**Questions For Authors:**

In early papers about network-based plan representations actions are sampled from a softmax distribution, parameterised by the network-calculated weights associated with applicable actions only. It would be interesting future work to extend verification to a wider range of network-based policies, such as these nonstationary policies. Has this already been done? I _guess_ that the SAT problem might be relatively easy to solve, because the solver would not have to resolve what action will take precedence as per the submitted manuscript.

**Reproducibility:**

4: Authors promise to release code and domains (whichever apply).

**Strengths Of The Paper:**

The paper is on a topic in a highly relevant area, and the empirically measured impact of the developed representational contributions, as evidenced in Column 11 of Table 1, is significant. The work is a short, detailed and meaningful practical contribution. I am not aware that these specific optimisations have been considered before for this problem.

**Weaknesses Of The Paper:**

The most substantial weakness was the incremental nature of this short study.

Experimental comparison with "NoApp"---i.e., wrong execution semantics---is strange, because as the authors observe such semantics are not comparable or faithful due to 'stalling'. I am not convinced this is a very useful addition to the experimentation.

---

> ### Author Rebuttal · Authors · 2024-01-25
>
> Thanks for the review.
>
> Concerning the comparison with NoApp:
> This comparison is needed to evaluate verification with vs without applicability-filtering, which is necessary to validate our claims that 1. a naive implementation of applicability-filtering incurs a huge performance loss, while 2. thanks to our enhancements, we approach the performance range of verification without applicability-filtering. The comparison has to be made with care as the underlying policy executions are not the same; but nevertheless the comparison meaningfully supports these two high-level claims.
>
> Questions:
> 1) We are aware of previous work on network-based planning policies (like ASNets, (Toyer at al. 2020)), that apply a masked softmax activation (masking out out inapplicable actions). If such a policy samples from the softmax distribution rather than choosing an argmax action, then the verification problem indeed changes. However a SAT nature still persists as we then need to determine which actions may have a softmax probability>0. And for applicability-filtering, that condition replaces the argmax condition we have now, which again results in a related problem. It remains future work to determine how to best deal with this and what kind of performance one gets.

---

### Meta-Review · Area_Chair_iakp · 2024-02-01

**Recommendation:** Accept (Oral)
**Confidence:** 5

**Metareview:**

Pros: The suggested representation optimizations are somewhat straightforward but result in a substantial performance improvement. The ablation study provides good evidence and some detailed analysis of the impact. The work is directly relevant to ICAPS.

Cons: The paper is a bit compressed - more reader-friendly explanations and diagrams are not included (and likely cannot be included due to space) - and this hurts the comprehensibility of the paper.

The reviewers' consensus on this submission (which I agree with) is that it is an appropriate use of a short paper and that the pros out-weigh the cons. We recommend that the authors strive for additional clarity (within the bounds of short paper) in their final version.

**Ethical Considerations:**

(1) Not Applicable: The paper does not have any ethical considerations to address